# CRAFTING LAYERED DESIGNS FROM PIXELS

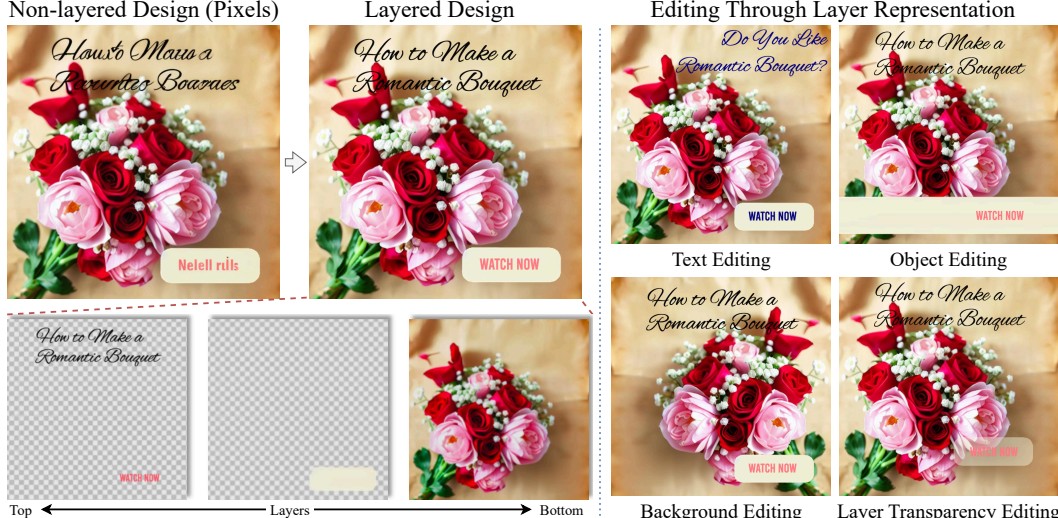

Figure 1: We generate layered designs from non-layered design reference images (in pixel format) by extracting background, objects, and text layers with optional further refinement. The obtained layered representation significantly eases the design process by facilitating a variety of layer-based editing operations. We also include a gallery for more visualizations in Appendix A.

## ABSTRACT

Graphic design plays a vital role in communicating ideas, values, and messages. During the design process, designers typically organize their work into layers of text, objects, and backgrounds to facilitate easier editing and customization. However, creating designs in such a format requires significant effort and expertise. On the other hand, with the advancement of GenAI technologies, high-quality graphic designs created in pixel format have become more popular and accessible, but with the inherent limitation of editability. Despite this limitation, we recognize the significant reference value of these non-layered designs, as human designers often derive inspiration from these images to determine layouts or text styles. Motivated by this observation, we propose **Accordion**, a graphic design generation framework built around a vision language model playing distinct roles in three key stages: (1) reference creation, (2) design planning, and (3) layer generation. Distinct from existing methods, by using the reference image as global design guidance, our approach ensures that elements within the design are visually harmonious. Moreover, through this three-stage framework, Accordion can benefit from an *unlimited* supply of AI-generated references. The stage-wise design of our framework allows for flexible configuration and various applications, such as directly starting from the later two stages given user-provided references. Additionally, it leverages multiple vision experts such as SAM and element removal models to facilitate the creation of editable graphic layers. Experimental results show that Accordion generates favorable results on the DesignIntention benchmark, including tasks such as text-to-template, adding text to background, and text de-rendering. Furthermore, we fully explore the potential of Accordion to facilitate the creation of design variations, validating its versatility and flexibility in the design workflow.

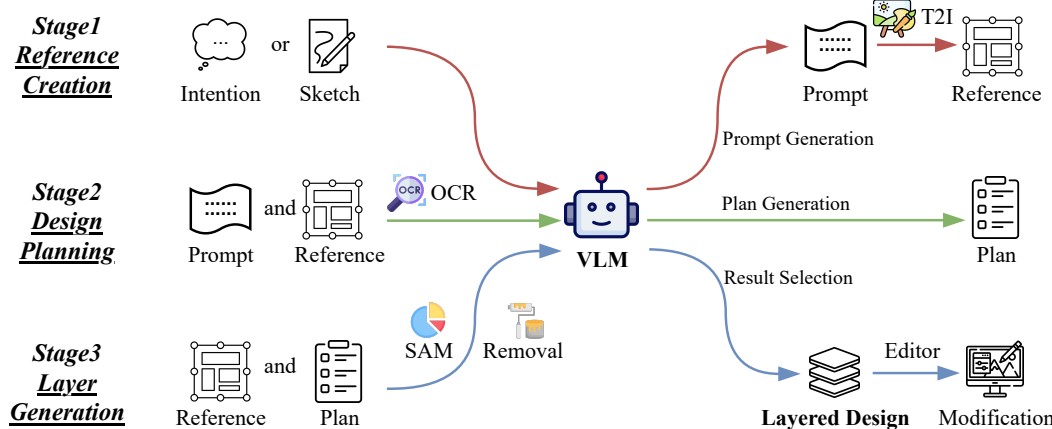

Figure 2: Overview of the proposed Accordion, which is built around a Vision Language Model (VLM). It consists of three stages for constructing the design reference, plan, and layers respectively. Multiple vision experts (*e.g.,* OCR and SAM) are employed to create layered designs.

# 1 INTRODUCTION

Graphic design is an important media format for modern visual communication. It boosts the clarity, aesthetic appeal, and communication effectiveness of digital content, impacting a wide array of real-world applications such as advertisement and user interface design Casner & Larkin (1989). Given the specialized application scenarios, graphic designs differ significantly from the natural images Deng et al. (2009). Specifically, graphic design is intrinsically constructed in a layered format, including distinct layers for text, foreground objects, and background. This structured layering allows for editability and customization, enabling designers to perform a variety of modifications, such as modifying text and backgrounds, thereby catering to specific user needs.

Despite the significant utility, creating layered graphic designs is a prohibitive task for most people due to the need for design expertise and huge effort. With the aid of image GenAI models, more design images have become available in rasterized pixel format. While they are visually compelling, they inherently lack editability. Even for simple operations such as horizontal flipping, text becomes unreadable since they are not separated from the background or other elements. Although users can employ some image editing tools Chen et al. (2023); Jia et al. (2024); Hertz et al. (2022) to modify the attributes of elements, such an approach is inconvenient compared to operations directly applied on the layer representation.

Nevertheless, we still believe rasterized designs are of great value for creating layered designs, by realizing that human designers naturally use rasterized designs from textbooks or other sources as references to get inspiration in their design workflows. For instance, designers will use them in the beginning to explore suitable layouts, deciding where to place objects and what style of typography to use to achieve visual harmony. Given this practice, we aim to leverage rasterized designs as references to create editable multi-layer designs, as illustrated in Figure 1.

Based on this motivation, we introduce **Accordion**[1], a framework as demonstrated in Figure 2 built around a vision language model (VLM) consisting of three stages: (1) *reference creation*, (2) *design planning*, and (3) *layer generation*, while the VLM plays different roles in these stages. Notably, the framework can leverage a diverse range of AI-generated references. It also offers the flexibility to start from the second stage when users explicitly provide designs as references. Besides, the VLM uses some vision experts in the design process. For example, SAM Kirillov et al. (2023) and removal models Rombach et al. (2022) are used for element extraction and background filling.

---

[1]Our method is named "Accordion" because it unfolds rasterized designs into layered designs, similar to how an accordion expands. Besides, the proposed Accordion framework harmoniously integrates each element within a layered design, much like orchestrating every note harmoniously in a musical score at the concert.

In contrast to earlier work of text rendering methods such as TextDiffuser Chen et al. (2023) that specifies the location of text before designing the background, and unlike COLE Jia et al. (2023) that starts with the background to design text in the visual domain, our method starts with a global reference image representing the target design as a whole. This initial reference image globally orchestrates the layout and visual properties of various elements, ensuring overall visual harmony and preventing the conflicts that may arise in the step-by-step generation where earlier design decisions may not work well with later ones.

Extensive experimental results and visualizations demonstrate the strong design capabilities of our method, especially its superior performance across various tasks on the DesignIntention benchmark Jia et al. (2023), including text-to-template, adding text to background, and text de-rendering. Additionally, we explore the potential of Accordion to facilitate creative design variation, including using upstream generative models to modify references, applying inference time variations, and utilizing downstream generative models for further variations based on the extracted layers. This shows the significant role of Accordion in the design process with its versatility and flexibility.

## 2 RELATED WORK

**Layered Design Generation**  There are a few investigations that focus on generating layered designs Jia et al. (2023); Inoue et al. (2024); Shimoda et al. (2021) considering the need for editability and customization. For instance, COLE Jia et al. (2023) starts from a brief user-provided prompt, employing multiple large language models (LLMs) and diffusion models to generate each element within the design. Even though COLE uses language to comprehensively plan the design, it still visually constructs the design step-by-step, starting from the background, then generating objects, and finally the text. This sequential approach may lead to visual conflicts such as failing to allocate sufficient or suitable space for text or objects when generating background, often resulting from the lack of a global visual impression in mind. Open-COLE Inoue et al. (2024) adheres to the architecture of COLE but incorporates certain simplifications, such as omitting the object generation stage. De-Render Shimoda et al. (2021) takes advantage of rasterized design images, leveraging several models to predict text attributes and extract background. Similar to Open-COLE, it does not explicitly separate out foreground objects. Additionally, commercial tools such as CanvaGPT canva (2024) utilize preset templates to adapt user prompts to new layered designs as noted in Jia et al. (2023). However, the diversity of designs is limited by the size of template libraries. We notice that a few methods Zhang & Agrawala (2024); Zhang et al. (2023c); Tudosiu et al. (2024) are designed for the layer generation of natural images. They are unsuitable for design images due to differences such as the absence of text layers in natural images.

Another line of research involves predicting the attributes of partial elements given any existing elements, which are then combined together to create a layered design. Some works focus on predicting the layout of elements within a design, typically emphasizing the importance of proper alignment and the avoidance of overlaps to enhance visual aesthetics Shabani et al. (2024); Jyothi et al. (2019); Li et al. (2019); Gupta et al. (2021); Hsu et al. (2023); Seol et al. (2024); Li et al. (2023; 2021); Zhu et al. (2024); Zheng et al. (2019); Zhou et al. (2022); Chai et al. (2023); Horita et al. (2024); Jiang et al. (2022); Cheng et al. (2024); Liang et al. (2024); Zhang et al. (2023a); Biswas et al. (2021); Yamaguchi (2021); Yu et al. (2022). Besides, some methods concentrate on predicting other attributes, such as text font and color, beyond layout within the design Inoue et al. (2023); Zhao et al. (2018); Lin et al. (2023); Biswas et al. (2024).

Overall, our method utilizes a VLM to predict attributes from global reference images for layered construction. It also employs visual experts to extract elements from the reference image to form layers, distinguishing it from existing methods.

**Non-layered Design Generation**  Existing GenAI methods are capable of creating an unlimited number of non-layered design images. Methods like Stable Diffusion Rombach et al. (2022); Podell et al. (2023); Esser et al. (2024), DALLE-3 Betker et al. (2023), and Ideogram ideogram (2024) have demonstrated this capability of design image generation guided by user prompts. However, these methods often suffer from text rendering errors Daras & Dimakis (2022), which can significantly impact the usability and aesthetic of the generated images.

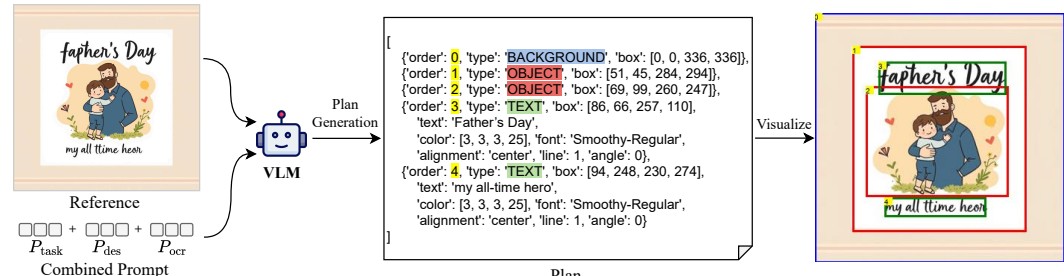

Figure 3: In the Stage2 design planning, VLM processes the reference image alongside a combined prompt to generate a comprehensive design plan. This plan includes detailed information about each design element, including the background, objects, and text.

To alleviate the text rendering issue, some works incorporate control over text areas during the image generation process Chen et al. (2023; 2024a); Tuo et al. (2024); Ma et al. (2024; 2023); Zhang et al. (2024); Zhao & Lian (2023); Ji et al. (2023); Liu et al. (2024a;b); Gao et al. (2023); Zhangli et al. (2024). While these methods improve the accuracy of text rendering, the final images are rasterized in pixel format, which prevents users from making edits easily, such as moving elements or adjusting text attributes. This limitation can reduce the practical usability of the generated graphics.

Overall, despite the inherent limitations of non-layered designs, we suppose these images are visually compelling and offer valuable references for the creation of layered designs.

## 3 METHODOLOGY

We sequentially detail the three stages of Accordion as illustrated in Figure 2, explaining the different roles the VLM plays in each stage. Note that we use GenAI images as references for illustrative purposes. Accordion can also start from the later two stages if users explicitly provide other types of references such as an existing design or a background layer.

### 3.1 STAGE1: REFERENCE CREATION

In this stage, our objective is to generate an image that serves as a global reference throughout the design process. Accordion is designed to accommodate a variety of inputs, including user-provided short intentions or sketch drafts, thus catering to diverse user preferences and needs. To facilitate this, we employ the VLM for **prompt generation**. As mentioned in COLE Jia et al. (2023), users may provide only short intentions for simplicity, such as "*create a poster for Father's day*". We use in-context learning by supplying the VLM with multiple examples for prompt enhancement following Open-COLE Inoue et al. (2024). This approach enables the VLM to generate detailed prompts such as *A father embraces his child in the center, surrounded by the text "Father's Day" and "My All Time Hero."*. It is observed that detailed and lengthy prompts can facilitate the creation of images with rich detail Chen et al. (2024b) and significantly improve text quality Chen et al. (2024a) for better reference. If users wish to provide more layout constraints, they can use sketch drafts. In such cases, the VLM is instructed to give detailed descriptions of the depicted objects, accurately depicting object positions and filling text in suitable areas. We show the prompt templates and visualizations in Appendix B. Subsequently, these generated prompts are fed into a text-to-image (T2I) model to generate images as references to be used in the next stage.

### 3.2 STAGE2: DESIGN PLANNING

In this stage, our objective is to derive a design plan based on the rasterized reference image. This plan should detail the placement of objects within the image to prepare for subsequent extraction, and provide the rendering attributes of text to facilitate the construction of text layers.

We employ VLM for **plan generation**. As illustrated in Figure 3, the VLM processes the reference image alongside a combined prompt, which includes a predefined task description $P_{task}$, a descrip-

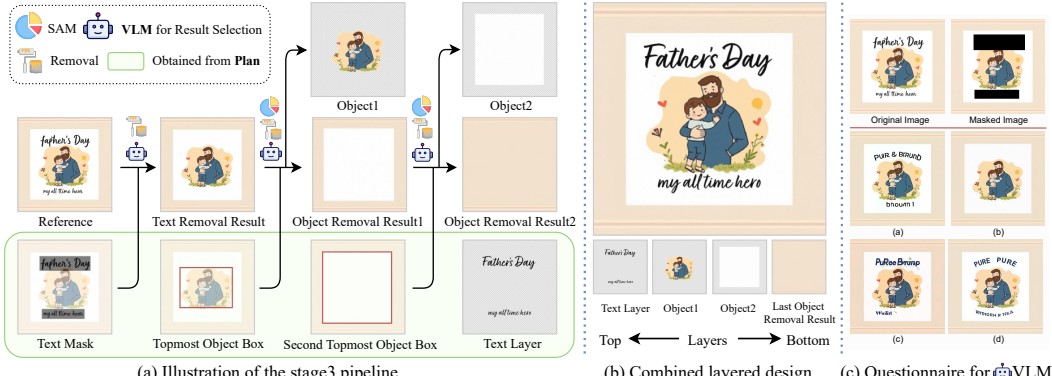

Figure 4: Overview of the Stage3 layer generation. (a) Guided by the plan, the reference image is processed first with text removal, and then with progressive foreground object extraction using SAM and an object removal model, obtaining the background image in the end. (b) The text, objects, and background are stacked into a layered design. (c) The VLM conducts result selection using questionnaires throughout this process for better removal results.

tion $P_{\text{des}}$, and OCR string $P_{\text{ocr}}$. The description is derived from the first stage detailed prompt, which is crucial for adapting textual content within the design plan, particularly for refining any nonsensical text produced by GenAI. For OCR string, we primarily utilize the text detection results of GenAI images, as the nonsensical content offers limited reference value. We show some cases of combined prompts in Appendix C.

As for the output, the VLM generates a sequence of dictionaries, each representing the attributes of elements in the image, arranged in a bottom-top order to facilitate further object extraction. The output includes bounding boxes for both background and foreground objects and detailed text attributes such as bounding boxes, content, color, font, alignment, line count, and angle. Box coordinates are normalized to the range [0, 336]. For the color attributes (R, G, B, A), we map the [0, 255] range to [0, 25] to facilitate learning inspired by COLE Jia et al. (2023). Extending these attributes is straightforward and can be accomplished by incorporating additional fields into the training dataset. So far, this design plan forms the foundation for the next layer generation stage.

## 3.3 STAGE3: LAYER GENERATION

In this stage, our goal is to construct the layered design based on the generated plan and reference image, as illustrated in Figure 4. Our approach is to extract and remove elements from the reference image, and then stack them back together to construct the final layered design.

Our initial step is to use a text removal model to erase text, recognizing that text regions are commonly placed on the top layer for enhanced readability. Based on the text removal result, our next focus moves to object removal, where we sequentially extract the topmost element according to the order outlined in the design plan. We employ the SAM, conditioned on the bounding box specified in the design plan, to extract the foreground object and obtain its corresponding mask. The mask and the intermediate image are fed into an object removal model to remove the foreground object. Notably, the object removal can be executed iteratively if multiple objects are detected in the design plan.

However, it is noticed that the removal model sometimes generates diverse results, not all of which are satisfactory. To ensure consistent quality, we design a questionnaire that enables the VLM to conduct **result selection**, as illustrated in Figure 4 (c). The top row shows the original image alongside the masked image, where the text is masked with text boxes for the text removal task, and the object is masked using the SAM segmentation map for the object removal task. During the training phase, we present the VLM with the ground truth of the removal alongside three generated removal results, training it to select the highest quality option. In the inference stage, we generate four removal results for the VLM to choose the best one. We showcase the task prompt and some samples in Appendix D.

Figure 5: For the same design, we employ three types of references for the VLM, including the original designs, designs with nonsensical text, and backgrounds without text.

Finally, all the extracted objects and the last object removal result serving as the background are combined with the text layer rendered from the design plan to create a layered design.

## 4 EXPERIMENTS

### 4.1 IMPLEMENTATION DETAILS

**Dataset.** We employ an in-house layered design dataset **Design39K**, with 39,233 samples for training and 492 samples for validation. This dataset comprises a diverse array of designs, including posters, book covers, advertisements, etc. Each sample is accompanied by a description. It is easy to extract layer information from these designs, including text, objects, and backgrounds, along with various attributes. We present more details about the dataset in Appendix E. As shown in Figure 5, to enrich the reference sources, we incorporate the following types of designs for training: (1) *Original designs*. We wish the model to conduct text de-rendering by directly parsing these original designs; (2) *Designs with nonsensical text*. We employ the Stable Diffusion 1.5 inpainting model Rombach et al. (2022) to inpaint text areas with inpainting strength randomly set between 0.5 and 0.7, leading to the generation of nonsensical text by the model. This range is selected because strength outside this range can lead to either insufficient or excessive inpainting changes, which hinder effective training. Please note that in some cases inpainting results may not strictly maintain the original text style, with potential variations in color and font. We consider this acceptable as it enables the model to use references while generating creative variations. The goal is to enable our model to effectively handle GenAI designs; (3) *Background without text*. By removing all text from a design and using only the background image as a reference, we challenge the model to add text in appropriate context and locations. We organize the elements of the design into a list of dictionaries and convert them into a string format for training the VLM. We employ the same training objectives for the three aforementioned reference types. Besides, to train the VLM for result selection, for both text and object removal tasks, we utilize the removal model to generate three different results. These results are then combined with the ground truth and randomly shuffled to construct the questionnaire dataset. In total, we have prepared 156,932 samples, consisting of 39,233 training samples for each of the three different types of references and the VLM questionnaire.

**Selection of vision expert models.** For the Text-to-Image (T2I) model, we utilize Flux for its proficiency in generating high-quality references. Additionally, we use PaddleOCR as the OCR tool. SAM Kirillov et al. (2023) and inpainting ControlNet Zhang et al. (2023b) for Stable Diffusion 1.5 Rombach et al. (2022) with the prompt "nothing in the image" are used in the layer generation stage. Importantly, these models are modular and can be replaced when more advanced alternatives are available. We detail each expert model in Appendix F.

**Training, inference, evaluation, and visualization.** We use the vision language model LLaVA-1.5-7B Liu et al. (2023) as the cornerstone of Accordion. The model is trained on the aforementioned 156,932 samples using LoRA Hu et al. (2022) with learning rate 2e-4 for 6 epochs, conducted on 8 × 80G A100 GPUs for 36 hours. The reference image is scaled with the longer side set to 336 pixels following Liu et al. (2023) for the VLM. The removal models operate at a resolution of $512 \times 512$, which is the size of the final output. During inference, the average generation time per sample is 43.7 seconds. For evaluation, we use the DesignIntention benchmark Jia et al. (2023) which provides 500 detailed prompts across various design domains. We visualize the layered design through a Streamlit HTML frontend, with details provided in Appendix G.

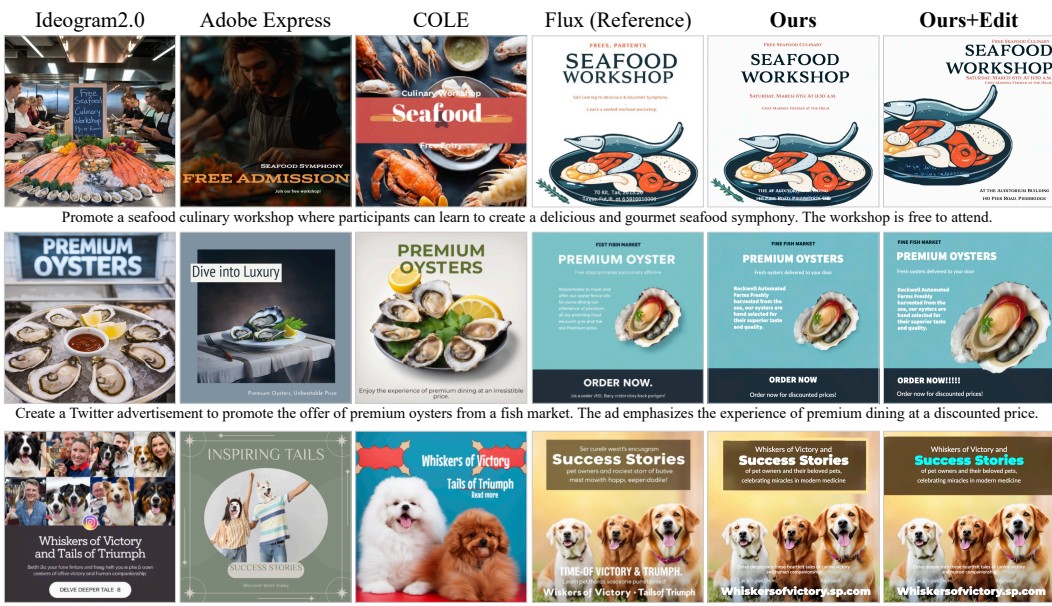

Figure 6: Qualitative results of text-to-template show that our method uses references generated by Flux to create layered designs where elements appear more harmonious, and supports layered editing compared with Ideogram2.0. Intentions are shown below each image. Best zoom in to view.

## 4.2 ABLATION STUDIES

In the following, we explore the efficacy of some architecture choices using the validation dataset.

**Should the VLM be trained separately or jointly across multiple tasks?** Here we investigate whether joint training gains benefits or leads to degradation. The results for each task are presented in Appendix H. We observe that joint training and separate training yield comparable average scores, with joint training slightly outperforming by a margin of 0.82%. We opt for a more compact model architecture and ultimately choose joint training.

**Does OCR prompt enhance VLM?** Here we assess the impact of incorporating OCR prompt. Considering the text recognition task for parsing the original design, we observe improvements in paragraph-level OCR Normalized Edit Distance (NED) by 7.23% (61.28% to 68.51%). Additionally, for the text detection task in both the original and GenAI designs, the average detection F1 score is improved by 5.46% (73.12% to 78.59%). So we use OCR prompt to enhance model performance.

**Does removal task benefit from VLM result selection?** We evaluate the efficacy of using questionnaires for the result selection process. The results indicate that for text removal tasks, the PSNR increases from 31.31 to 31.97, and for object removal tasks, it improves from 29.33 to 29.59. It demonstrates the effectiveness of this approach in enhancing the results of removal tasks.

## 4.3 EXPERIMENTAL RESULTS

**Text-to-template.** We generate reference images using Flux flux (2024) and compare them with commercial tools such as Ideogram2.0 and Adobe Express, as well as with images generated by COLE Jia et al. (2023). Some samples are shown in Figure 6. It is noteworthy that although the images produced by Ideogram2.0 appear fancy, they lack editability, and some text areas exhibit artifacts. Adobe Express produces vector designs with professional styles, but the generated text does not always match the input query closely. For example, in the second row, we intend to generate an advertisement for an "oyster discount", but the output contains a text saying "dive into luxury". COLE can generate layered designs, but the elements lack visual harmony. For example, in the first row, the placement of "Culinary Workshop" and "Free Entry" looks unnatural; and in the third row, the positioning of two red objects conflicts with the foreground text. Our method uses Flux to create

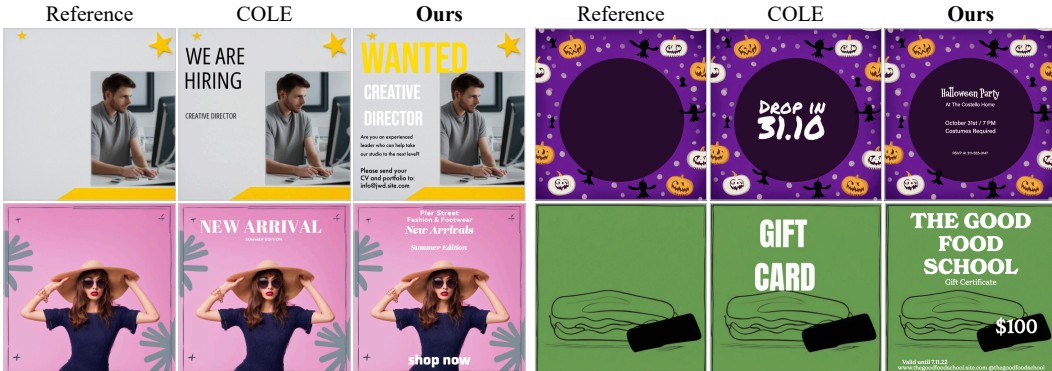

Figure 7: Qualitative results of adding text to background show that our method not only places text harmoniously in terms of style and layout but also effectively utilizes space to create more complex and aesthetically pleasing designs than COLE.

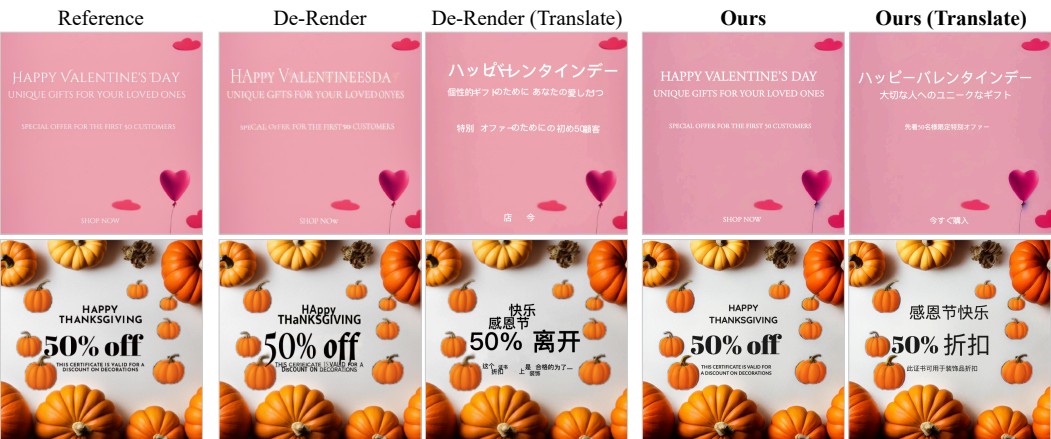

Figure 8: Qualitative results of text de-rendering. Our method groups text at the paragraph level to predict their attributes, thereby enhancing visual harmony. Through the translation task, our method better maintains semantic coherence, style harmony, and spatial alignment compared with the word-level approach De-Render.

references. We observe that due to the exceptional generative capabilities of Flux, the rasterized images it produces feature visually harmonious elements, despite the presence of nonsensical text. Then our method extracts objects from these references, learns the text style, and improves upon nonsensical text areas in the references, demonstrating superior performance. We also showcase editing results in the last column, including operations on text and objects.

We validate the quantitative results using the protocol provided by COLE, which evaluates performance across five metrics: (i) design and layout, (ii) content relevance, (iii) typography and color, (iv) graphics and images, and (v) innovation. These metrics are scored via preset prompt queries to GPT-4V, with scores ranging from 0 to 10 and higher scores indicate better performance. Our comparisons are primarily with academic methods following Inoue et al. (2024) because commercial tools typically lack open-source code or accessible APIs for large-scale testing. As shown in Table 1, our method achieves the highest average score (6.5) compared to existing layered approaches such as COLE and Open-COLE. Notably, we specifically evaluate the content relevance of images generated by Flux and our method. The experimental results show that our method outperforms Flux by 0.4 in this metric (7.0 for Flux *v.s.* 7.4 for ours). This improvement demonstrates that our method refines the image by replacing nonsensical text with relevant information, thereby enhancing the content relevance of text layers to backgrounds.

Table 1: Quantitative results for the comparison of layered design methods using the DesignIntention benchmark. Five metrics include: (i) design and layout, (ii) content relevance, (iii) typography and color, (iv) graphics and images, and (v) innovation. Our method achieves the highest average score.

| Layered Design Methods | (i) | (ii) | (iii) | (iv) | (v) | Avg. | Remark |
|---|---|---|---|---|---|---|---|
| COLE Jia et al. (2023) | 6.0 | 6.9 | 5.7 | 6.2 | 5.1 | 6.0 | - |
| Open-COLE Inoue et al. (2024) | 6.3 | 7.0 | 5.6 | 7.1 | **5.3** | 6.3 | No object layers |
| **Accordion (Ours)** | **6.7** | **7.4** | **6.1** | **7.3** | 5.1 | **6.5** | - |

**Adding text to background.** Since COLE provides the SVG-format results for the DesignIntention benchmark Jia et al. (2023), we can easily remove the text to obtain the background. As illustrated in Figure 7, we compare our method with the results from COLE. The results demonstrate that our method achieves good harmony in both layout and text style. In addition, we observe that COLE tends to produce layouts that are relatively simple, often avoiding longer sentences. Through analysis, it is evident that our model is able to generate text with more informative content. Our samples show a greater overall text length, averaging 61.7 characters compared to 42.3 characters for COLE (approximately 1.5 times longer). In other words, our approach can fully utilize empty space in the background to increase information density and enhance overall aesthetics. This makes our model especially well-suited for scenarios like detailed reports and comprehensive advertisements, where it is crucial to convey complex information effectively.

We quantitatively compare our results with COLE using GPT-4V, where images generated by both methods are concatenated horizontal and shown to GPT-4V to assess which one has higher quality. The task prompt is provided in Appendix I. The experimental results demonstrate a preference for our model on 52% of all the samples compared to 48% for COLE. Notably, this preference is achieved despite our model being trained on a considerably smaller dataset. Our model is trained using only 39K data, substantially fewer than the 100K used to train COLE.

**Text de-rendering.** In Figure 8, we validated the text de-rendering capability of our model in comparison with De-Render Shimoda et al. (2021). Note that our method treats paragraph-level text as a single entity. One advantage of paragraph-level representation is the uniformity of style, a feature that De-Render struggles to achieve. In contrast, De-Render predicts a style for each word individually, which can lead to a visually disorganized appearance. Moreover, we showcase our advantage with a translation application. By grouping words at the paragraph level, our approach effectively considers the coherence of sentence semantics during translation, ensuring consistent style and alignment of adjacent words. In contrast, since De-Render operates at the word level, it lacks overall contextual information in translation application, leading to disorganized layouts with overlapping text. Please note that since our method operates at the paragraph level, we cannot guarantee that each line of text will align exactly as in the original. However, we believe this is acceptable. As the text is already separated into layers, adjustments can be made easily.

**Design variation creation.** We demonstrate six methods based on our approach for creating design variations under three categories: *variation by upstream models*, *variation at inference time*, and *variation by downstream models*. As shown in Figure 9, sub-figures (a) and (b) employ an upstream image variation model Xu et al. (2023) and an inpainting model Podell et al. (2023) respectively to generate diverse reference images. Typically, generative models that produce text image variants usually produce nonsensical text. Our approach effectively refines these designs. For inference time variations, sub-figure (c) achieves variation by conditioning on a prefix and using different seeds during inference. Sub-figure (d) employs the reference switch. Specifically, after the text removal stage, it does not continue to follow the original text reference but instead uses the background as the new reference, adding text on the background to explore new text layouts. For the utilization of downstream generative models, sub-figure (e) utilizes a downstream layout generation model Visual Layout Composer Shabani et al. (2024) to create variations based on extracted layers. Sub-figure (f) uses Instruct Pix2Pix Brooks et al. (2023) to modify the background image. Direct application to the entire flattened image often results in artifacts, especially in regions with small text. We apply the editing only on the background layer and then recompose all the layers so that the quality of the text area is preserved. These methods demonstrate the versatility, flexibility, and critical role of our approach in the design workflow.

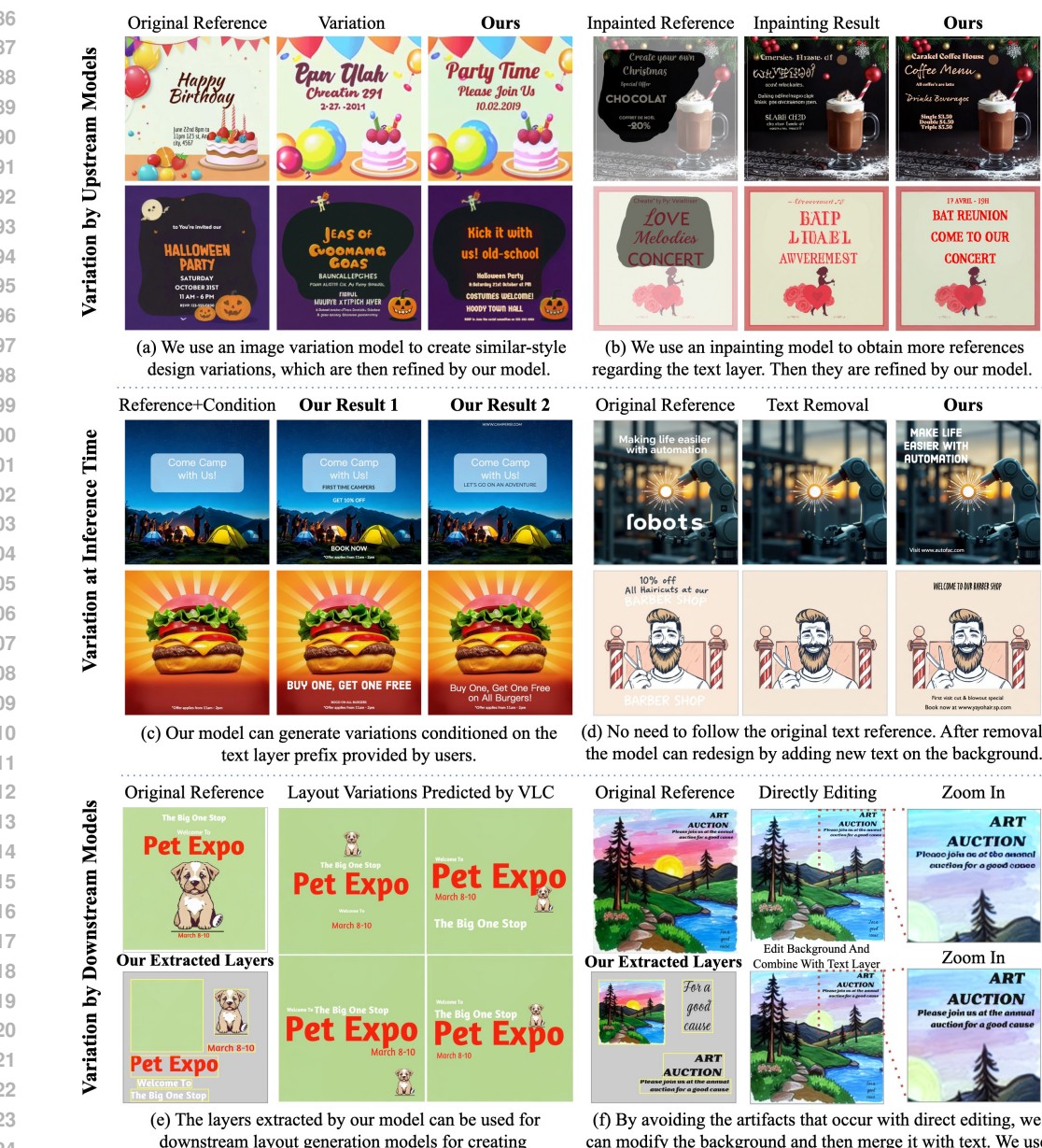

Figure 9: Six different methods are used to achieve design variation: (a) and (b) modify the reference image using upstream models; (c) and (d) create variations during the inference time; and (e) and (f) use downstream generative models based on extracted layers to generate further variations.

## 5 CONCLUSIONS

In this paper, we introduce the Accordion, a three-stage framework that leverages reference images to create layered designs. It exhibits superior performance on the DesignIntention benchmark compared to existing methods. Additionally, numerous applications underscore its flexibility, versatility, and controllability. For the *limitation*, we observe a small probability (approximately 2%) where the VLM-generated results cannot be parsed correctly. Additionally, about 3% of the font strings do not match any fonts in our database. Hence, some post-processing is needed to refine the result format. For *ethical concerns*, so far we have not identified issues. For *future work*, we aim to enhance the model's capability to create new backgrounds and objects based on references.

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

# Appendix

## A   GALLERY FOR MORE TEXT-TO-TEMPLATE RESULTS

We demonstrate more text-to-template results in Figure 13. Due to the text being layered and vectorized, operations such as selection and copying are feasible.

## B   PROMPTS FOR THE STAGE1 REFERENCE CREATION

Below we use in-context learning to make VLM expand the given prompt:

> Your task is to expand the original prompt into a detailed one. I will give you some examples.
>
> [Input 1] Create an advertisement for a fish market with a special offer of a 20% discount on seafood.
>
> [Output 1] A bustling fish market under a vibrant morning sky. Local vendors display an array of fresh, glistening seafood, from ruby-red lobsters to iridescent, silver fish. A large, colorful banner hangs overhead, proudly announcing a special offer with 20% discount on all seafood. The air is thick with excitement and the irresistible aroma of the ocean.
>
> [Input 2] Create a business card for a flower shop with a focus on blue tulips.
>
> [Output 2] An elegant business card lying on a white marble surface. The card is adorned with a captivating watercolor illustration of rich, azure blue tulips, their petals opening up to reveal layers of deep and light shades of blue. The shop's name is written in a sophisticated cursive font at the center, while contact details are subtly placed at the bottom right corner.
>
> Now based on the given prompt "Design a cutting-edge logo for a real estate agency named Golden Home.", please expand it into a detailed one.

When users wish to provide additional constraints, they can input sketches to the VLM. We employ the following task prompt template to query the VLM:

> You will be provided with a sketch that you need to analyze and describe meticulously, paying close attention to each detail depicted. Identify and describe where each object is located within the sketch. Note that the "xxx" symbols on the image are placeholders for text, which you should replace with appropriate content. Your description should capture the layout and the thematic elements of the design. As a reference information, this image is about "eating more apples is good for your health".

This template explicitly describes the task and indicates that specific text should replace the placeholder 'xxx'. It also includes a brief user intention, providing the VLM with an overview of the image. We demonstrate some cases in Figure 10.

## C   COMBINED PROMPTS FOR THE STAGE2 DESIGN PLANNING

We showcase some examples in the below. For training GenAI design, we use the following template:

> Parse and refine the attributions of text. Parse the objects, and backgrounds in the graphic design image. The caption of the image is The "Red White Bold Type" beverage label is a striking visual feast, designed to capture the essence of boldness and purity. With a vivid red and pristine white color scheme, the label features bold, assertive typography that commands attention. This design not only reflects the vibrant and robust flavors of the beverage but also appeals to consumers with its clean, contemporary aesthetic, making it a standout choice on

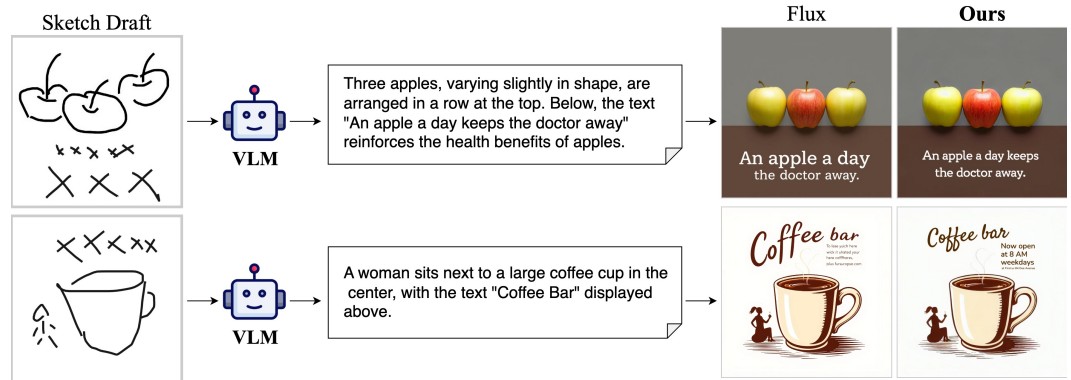

Figure 10: The VLM can convert the sketch draft into detailed prompt to generate references.

any shelf. Support OCR results are: [[(22, 64, 228, 132)], [(21, 126, 311, 211)], [(82, 208, 119, 215)]].

Parse and refine the attributions of text. Parse the objects, and backgrounds in the graphic design image. The caption of the image is The Facebook page cover for a modern record store should be a vibrant and engaging visual that encapsulates the essence of music and contemporary design. It might feature a collage of iconic album covers, interspersed with sleek, modern graphic elements that convey the store's cutting-edge aesthetic. Support OCR results are: [[(214, 89, 299, 120)], [(41, 86, 110, 138)], [(18, 121, 59, 176)], [(195, 121, 317, 147)], [(209, 175, 310, 197)], [(224, 197, 290, 219)], [(84, 219, 106, 237)], [(215, 232, 300, 246)]].

For training the original design, we use the following template. Please note that here we do not incorporate the description since the text within the design already contains massive information. Meanwhile, we integrate the OCR recognition result in the OCR string.

Parse the attributions of text, objects, and backgrounds in the graphic design image. Support OCR results are: [['THE COOD', (85, 15, 228, 51)], ['CREATIVE', (88, 51, 232, 85)], ['STUDIO', (84, 83, 196, 120)], ['2701Willow', (85, 218, 158, 236)], ['Charles,', (83, 235, 135, 253)], ['aneLake', (122, 228, 177, 243)], ['(555)555-0100', (85, 265, 174, 282)], ['@thegoodstudio', (86, 297, 180, 312)], ['www.thegoodstudio.site.con', (85, 310, 237, 324)]]

Parse the attributions of text, objects, and backgrounds in the graphic design image. Support OCR results are: [['CLEARANCE', (19, 213, 318, 255)], ['SALE', (14, 256, 136, 297)], ['2701WillowOaks', (203, 272, 300, 287)], ['Lane Lake Charles,LA', (203, 284, 321, 298)]]

For training the backgrounds with text, we use the following template. Note that the OCR string is omitted since there is no text within the design.

Add text on the background. And parse the overall graphic design. The caption of the image is floral green and pink wellness institute business card.

Add text on the background. And parse the overall graphic design. The caption of the image is The logo for Green Saw Carpenters captures the essence of the brands commitment to sustainable building practices and skilled craftsmanship. It features a stylized green saw blade,

intricately designed to resemble both a leaf and a carpentry tool, symbolizing the fusion of nature and construction.

# D    MORE DETAILS ABOUT THE QUESTIONNAIRE FOR RESULT SELECTION

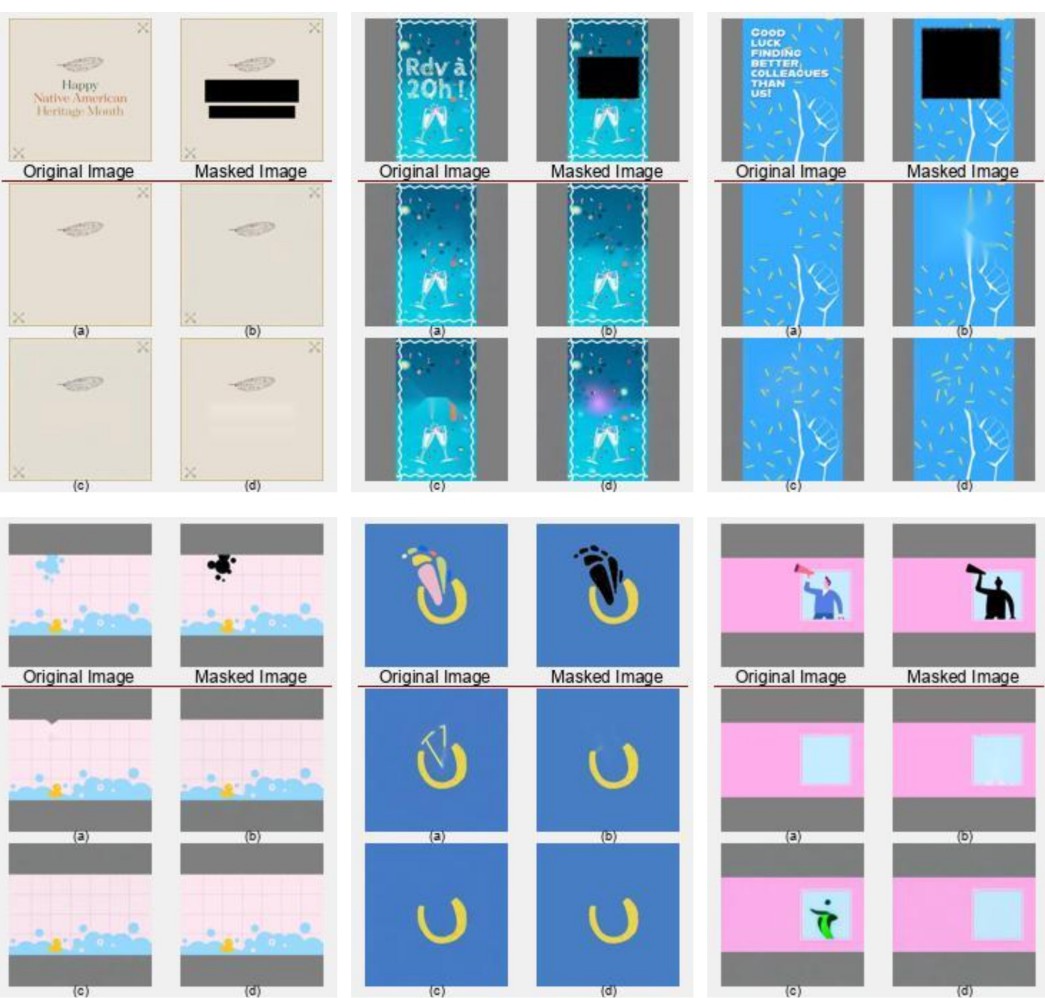

Figure 11: More examples about the questionnaire. The samples in the first row is for text removal, and the samples in the second row is for object removal.

We show the task prompt in the following, and display some cases in Figure 11.

The provided image appears to show four different results of a graphic design removal task. The first row displays the original image on the left and the masked image on the right. The second and third rows exhibit the corresponding outcomes of the graphic design removal. To evaluate the effectiveness of the results, the key criteria are: 1) the overall harmony and coherence of the image, 2) the purity and cleanness of the background, and 3) the absence of any additional, extraneous elements. Based on these criteria, please select the option (a, b, c, or d) that represents the best result.

## E    MORE DETAILS ABOUT DESIGN39K

Here we provide additional statistics about the in-house Design39K dataset. The average output sequence length is 728.32. To ensure that the majority of cases do not exceed this limit, we have set the maximum output length for the VLM to 1,536. Each design, on average, contains about 1.02 objects and 3.11 text regions. Notably, the dataset utilizes the title of each design as the description.

## F    MORE DETAILS ABOUT EACH VISION EXPERT

**Flux.** We use the open-source Flux.1 schnell to create the reference. We observe that, even without explicit character guidance like other methods Chen et al. (2024a); Tuo et al. (2024), Flux still synthesizes high-quality design references. Flux demonstrates robust performance in generating reference images compared with previous methods such as SD 1.5 Rombach et al. (2022). In some cases, Flux may struggle to generate design images according to the prompts, instead producing backgrounds without text. In these situations, users may need to attempt multiple times to obtain the desired reference image. The output size of the model is $1024 \times 1024$. The number of sampling steps is set to 4, and the maximum length of the input prompt is set to 256, both as default.

**PaddleOCR.** PaddleOCR is an open-source optical character recognition toolkit that provides practical and efficient solutions for text detection and recognition across various images. It has been observed that PaddleOCR also exhibits strong detection performance on GenAI images.

**SAM.** Segment Anything Model Kirillov et al. (2023) is an advanced segmentation model designed to perform highly accurate and versatile image segmentation across an extensive array of objects and scenes, enabling detailed and automated analysis of visual data. Here we employ the detection box to obtain the segmentation mask. Specifically, we employ the "sam-vit-base" architecture to get the segmentation mask.

**Removal model.** We achieve object removal results using the ControlNet inpainting model based on SD 1.5 Rombach et al. (2022), employing the prompt "nothing in the image" to erase specific content. The input size and the output size are $512 \times 512$. While we notice that a few samples exhibit color shifting, we consider this acceptable as the results still appear harmonious.

## G    DETAILS ABOUT THE STREAMLIT FRONTEND

We develop a Streamlit-based front-end system to facilitate the presentation and manipulation of layered graphic designs. This system enables the separation and individual rendering of various design elements, including text, images, and objects, thereby allowing for flexible control and real-time previewing of design components. Specifically, we leverage HTML and CSS to render text elements and employ the st.elements.image component to display images and objects.

## H    DETAILS ABOUT THE EXPERIMENT ON THE BENEFITS OF JOINT TRAINING

In Table 2, we present the evaluation results using various types of images as inputs. In Table 2, we present the evaluation results using various types of images as inputs. Note that we use normalized edit distance (NED) for evaluating OCR accuracy, which is particularly effective when our OCR is applied to paragraph-level long text following Chng et al. (2019). For color accuracy, we consider the predictions correct only when the values for red (R), green (G), blue (B), and alpha (A) channels are exactly accurate. When the input images are designs with nonsensical text or backgrounds without text, it becomes challenging to assess text-related metrics. Therefore, we rely solely on object-related metrics for evaluation. The results indicate that the scores for both categories are comparable. Given our goal to develop a compact model, we ultimately opt for joint training.

## I    TASK PROMPT FOR EVALUATING ADDING TEXT TO BACKGROUND

Here we use the following prompt for GPT-4V. The horizontal concatenation is demonstrated in Figure 12.

Here you will see two designs with the same background but different text content and place-ment. Please consider the factors such as layout, content relevance, typography and color scheme. Select which one is better, you can answer the left one is better / the right one is better, then detail the reasons.

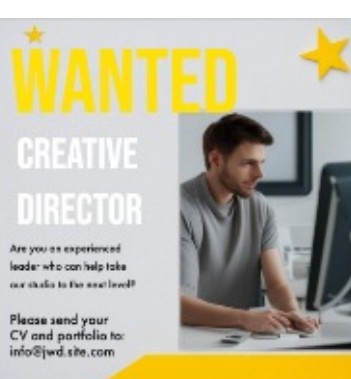

Figure 12: Samples generated by COLE and ours are concatenated for comparison.

Table 2: Ablation studies about the experiment on the benefits of joint training.

| Metrics | Separate Training | Joint Training |
|---|---|---|
| ***Original Design*** | | |
| Text Detection F1 | 75.42 | **78.59** |
| Text Recognition NED | **72.87** | 68.51 |
| Object Detection F1 | 82.17 | **84.64** |
| Color Accuracy | 26.66 | **28.09** |
| Font Accuracy | **24.51** | 21.62 |
| Line Number Accuracy | **86.96** | 86.28 |
| Alignment Accuracy | 87.28 | **88.60** |
| Angle Accuracy | 90.16 | **91.52** |
| ***Designs with Nonsensical Text*** | | |
| Object Detection F1 | 79.27 | **83.06** |
| ***Backgrounds without Text*** | | |
| Object Detection F1 | 83.52 | **86.94** |
| ***Questionnaire Result Selection*** | | |
| Selection Accuracy | **83.54** | **83.54** |
| ***Average Score*** | 72.03 | **72.85** |

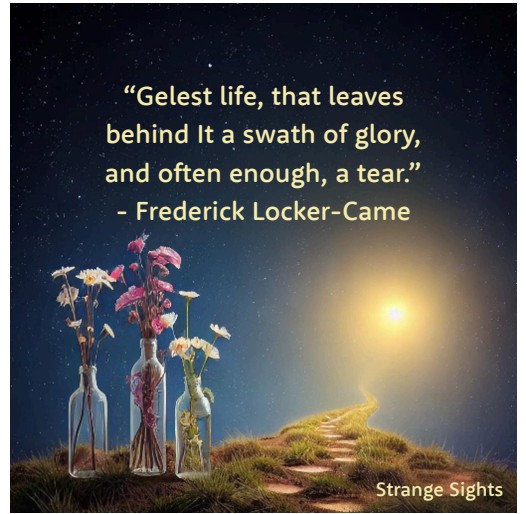
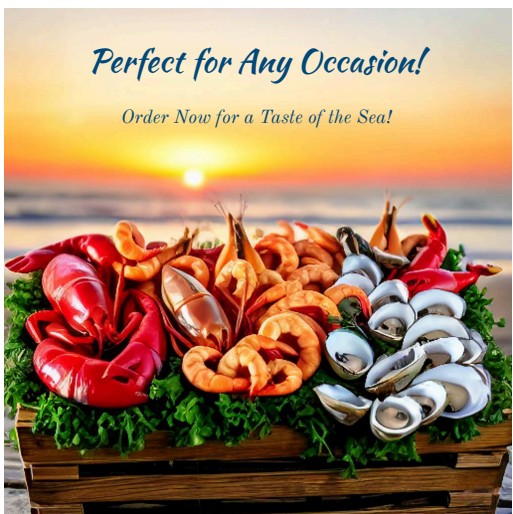
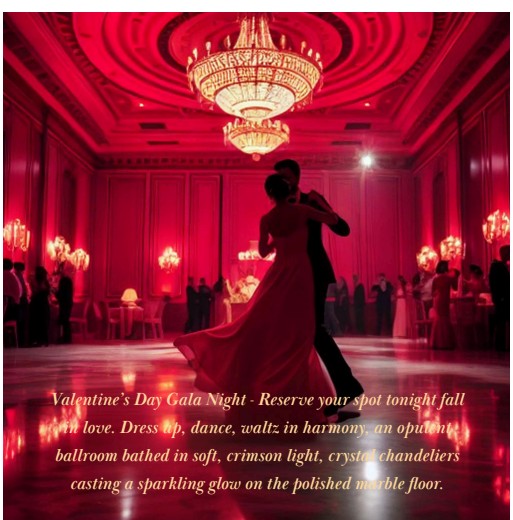
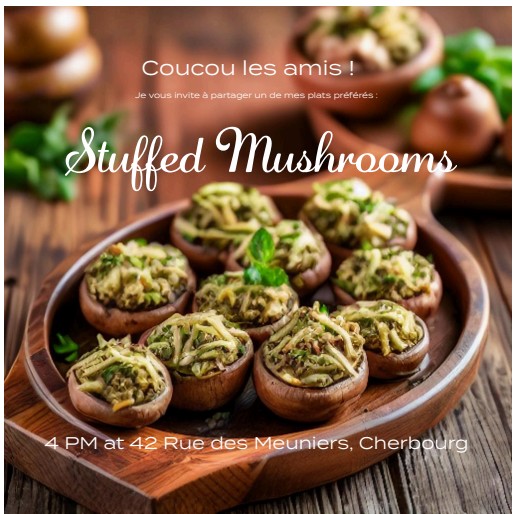
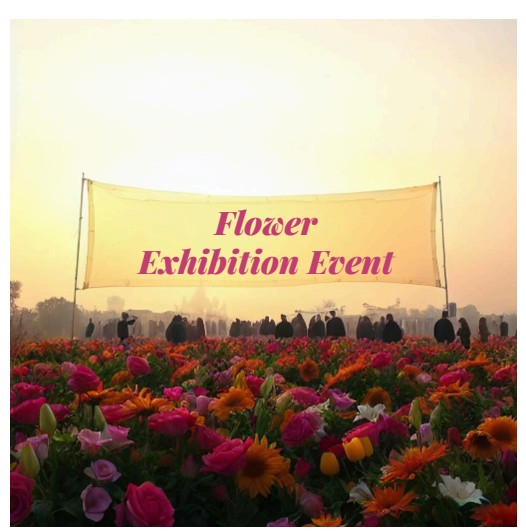
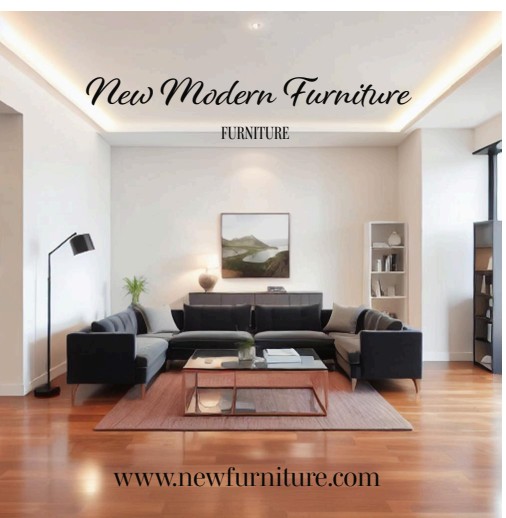

Figure 13: Visualizations of more text-to-template results.

