# OpenReview forum: "Crafting Layered Designs from Pixels"
_ICLR.cc/2025/Conference — ICLR 2025 Conference Withdrawn Submission_

### Official Review · Reviewer_VF8x · 2024-10-31

**Soundness:** 3
**Presentation:** 3
**Contribution:** 2
**Rating:** 3
**Confidence:** 4

**Summary:**

Based on existing Vision Language Model, the paper presented a layers-aware graphics design system, named Accordion. The system can be used for reference creation, design planning, and layer generation. The pipeline is comprised of three stages: reference creation, design planning, and layer generation. The experiments demonstrate applications of text to template, adding text to background, text de-rendering, and design variation creation.

**Strengths:**

The system is comprised of various existing components, and seems to work well. It is able to ease the graphics design work.

The result images are delightful. I hope the system could be accessed publicly in the future.

**Weaknesses:**

The innovation is limited. The work integrates various existing tools and large models, such as SAM, VLM, to compose the system. In the Section 3 (Methodology), there is no specific content describing the work, but there is only high-level summary. Hence, I cannot find enough specific academic contribution.

This is an engineering work. I consider it is not suitable for an ICLR research paper.

**Questions:**

I'd like to see the runtime table for Stage 1, 2, 3 and various editing tasks, as waiting time highly has a big impact on an editing system.

Is the system adaptable to different resolution images. If given a high resolution image (>5M pixels), how about the running performance.

**Details Of Ethics Concerns:**

The system may be used to generate fake images.

---

### Official Review · Reviewer_rr7V · 2024-11-01

**Soundness:** 2
**Presentation:** 2
**Contribution:** 2
**Rating:** 5
**Confidence:** 3

**Summary:**

This paper proposes some components for visual designs. It has stages for layer extraction, background processing, element planning, etc, to help the design generation.

**Strengths:**

The problem and research direction seem solid and has many real-world applications. Some recent commercial products like ChatGPT Canvas also validate the importance of these research directions.

The paper has shown many results with different application types.

**Weaknesses:**

It is not very clear which part is proposed by this paper: The Reference Creation is an existing Vision LLM. (This paper proposed a prompt that works well for it). The OCR, background removal, SAM, inpainting are all existing models. The Vision LLM is trained with an internal dataset Design39K, which seems the contribution of this work. But this contribution (presenting a finetuned VLM) seems relatively weak.

**Questions:**

I think this paper may need a major revision to make the exposition much clearer. The authors may want to show at the very beginning that the intended use case of this application is mainly converting AI-generated design images into real design files with layers and texts, as a post-processing for image generators like Flux and SD.

Currently the first stage of this method is “reference creation”, making readers to think that the references are outputs. But the intended application is mainly using existing reference as inputs.

---

### Official Review · Reviewer_RPLH · 2024-11-02

**Soundness:** 3
**Presentation:** 3
**Contribution:** 2
**Rating:** 5
**Confidence:** 3

**Summary:**

This paper presents a workflow for generating layered graphic designs using visual language models (VLMs). The workflow involves three stages from generating reference, design planning, and layers. The proposed workflow can facilitate creation of design variations.

**Strengths:**

* the overall workflow successfully generate nice layered graphic design.
* the workflow is versatile for different design scenario
* the proposed method is straightforward as long as the training data is provided.

**Weaknesses:**

* Although the results seem better than other compared method, it is not clear whether this is because the better image generative model. I recommend to have a fair comparison, e.g., applying the planning and layering method to the design image generated by other image generative models.
* the novelty of the method seems limited. I think the main contribution is the process of generating design planning before layering decomposition. But I think it lacks the comparison on combining different previous works with the proposed unified workflow.
* lack of experiment and results on applying the proposed method on graphic design in the wild.

**Questions:**

* it is still very unclear to me why the method need to generate the reference image in the first place instead of using an existing deisgn?
I think it makes the overall proposed workflow like a system or a product, without a concentrated technical contribution.
* I think the second and the third stages can be achieved by combining other existing works. Therefore, I want to ask is there any specific reason why those existing works for text de-rendering and layer decomposition or code generation from a given design cannot work?

---

### Note · Authors · 2024-11-13

**Comment:**

We appreciate all the reviewers' efforts and valuable feedback on our manuscript, and decided to withdraw the manuscript.

**Withdrawal Confirmation:**

I have read and agree with the venue's withdrawal policy on behalf of myself and my co-authors.